# Facial AU-aid hypomimia diagnosis based on GNN

**Yingjing Xu**[1]                                                                 POPPYXU@ZJU.EDU.CN

**Bo Lin**[2,3] *                                                             RAINBOWLIN@ZJU.EDU.CN

**Wei Luo**[4]                                                               LUOWEIROCK@ZJU.EDU.CN

**Shuiguang Deng**[2]                                                              DENGSG@ZJU.EDU.CN

**Jianwei Yin**[2]                                                          ZJUYJW@CS.ZJU.EDU.CN

[1] *School of Software Technology, Zhejiang University*

[2] *College of Computer Science and Technology, Zhejiang University*

[3] *Innovation Centre for Information, Binjiang Institute of Zhejiang University*

[4] *Second Affiliated Hospital, Zhejiang University School of Medicine*

## Abstract

Hypomimia is a prevalent symptom of Parkinson's Disease(PD). It is characterized by reduced facial expression and delayed facial movement. The work proposes a framework to use Graph Neural Network(GNN) to extract related action unit(AU) features on the facial smiling videos to help to improve the recognition of hypomimia with PD. AU is an effective representation of the facial state and movement, while GNN has great capability to present relationship information between facial areas. A related AU representation can pay more attention to the relationships between the facial areas in order to increase the accuracy of the diagnosis. Experiments were conducted using an in-house dataset of 105 facial smiling videos, which contains 55 healthy control(HC) participants and 50 PD patients. Our method's performance was compared to that of random forest (RF) and support vector machine (SVM) classifiers. Our method achieved an Accuracy, PPV, TPR, and F1 score of {**91.7%, 92.8%, 90.6%, 91.7%**}, while the RF and SVM achieved {84.5%,84.8%, 82.7%, 83.7%} and {88.7%, 88.0%, 88,7%, 88.3%} respectively on the dataset.

**Keywords:** Hypomimia, Parkinson's Disease, Action Unit, GNN

## 1. Introduction

Parkinson's disease is a common neurological disease, the prevalence of Parkinson's disease in the population over 65 years old is about 1.7%. Facial hypomimia is one of the manifestations of motor symptoms, the patient's facial expression ability is impaired, and the delay of facial movement leads to the reduction of facial movement. The MDS Unified-Parkinson Disease Rating Scale(MDS-UPDRS) is an authoritative scale used to assess PD in the clinic. With the development of facial recognition, Action Unit(AU), a technique to represent and quantify facial status, has been widely used and can effectively reflect facial movement. In this paper we propose a video-based hypomimia recognition framework that utilizes AU that combing facial area information and uses GNN to measure the relation between facial AU areas. Our method can extract comprehensive AU information and outperform traditional machine learning methods in the experiment.

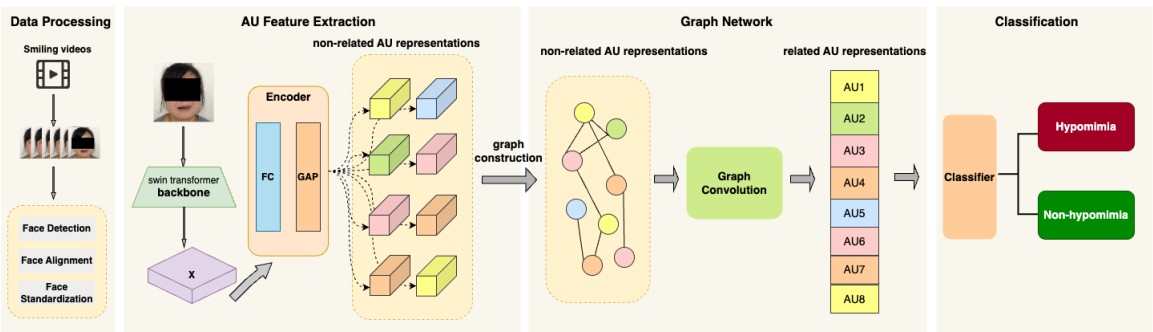

Figure 1: The pipline of proposed method.

## 2. Methods

We applied a AU intensity prediction method(Luo et al., 2022) in our method. As detailed below, Figure 1 depicts the pipeline of the proposed method. First, Smiling videos are converted into aligned frames after dataprocessing. Next, 8 AU representations are extracted by Swin Transformer(Liu et al., 2021) . After graph construction and convolution, we can get related AU representations. At last, classifier determines hypomimia by related AU representations.

**Data processing.** The original dataset is smiling videos from HC participants and PD patients. The videos were recorded at 1920×1080 pixels with 60 frames per second. For each frame, to avoid the interference factor of head position and lighting intensity, we performed face detection, face alignment, and face normalization sequentially by using MTCNN(Zhang et al., 2016) .

**AU feature extraction.** To extract the full face representations, we used the Swin Transformer backbone(Liu et al., 2021) . The encoder contains a fully connected layer(FC) and a global average pooling layer(GAP). 8 non-related AU representations can be extracted from a full face representation by the encoder.

**Graph convolution.** We defined the non-related AU features extracted by Swin Transformer as nodes of the graph, and the similarity between pairs of AUs calculated by K(K=2) nearest neighbors algorithm was defined as edges of the graph. We then performed graph convolution on the graph and obtain representations containing related AU information.

**Classification.** The classifier was a fully connected neural network, and it used cross-entropy loss to determine hypomimia.

## 3. Expermental results

**Experimental setup.** We collected 105 smiling videos from The Second Affiliated Hospital of Zhejiang University, which included 50 smiling videos from PD patients and 55 smiling videos from HC participants. After a series of processing by MTCNN, the smiling videos were split into training, validation, and test sets according to each person using Hold-out Method, which were divided into 60, 20, 20 people respectively. The corresponding video

---

* Corresponding author

frames were 20,479, 6,898, and 8,030 frames respectively. Support Vector Machine(SVM) and Random Forest(RF) is used as the baseline of the experiment, in which the input of baseline is AU intensity values. The evaluation metrics used were accuracy, positive predictive value(PPV), True positive rate(TPR), and F1 score. For hyperparameters, the learning rate was set to 0.001, the batch size is set to 24, and the epoch is set to 20.

**Results.** The proposed method has shown promising results in extracting facial expressions and identifying hypomimia. As shown in Table 1, the results showed that our method achieved the best performance, with an accuracy of 0.9167 and an F1 score of 0.9170 on the test set. In comparison, SVM achieved an accuracy of 0.8869 and an F1 score of 0.8834, while RF achieved an accuracy of 0.8448 and an F1 score of 0.8373. Our method outperformed the traditional classifiers in terms of accuracy, PPV, TPR, and F1 score, indicating that the graph representation of facial expressions can better capture the relationship between facial areas and improve the diagnosis of hypomimia with PD.

Table 1: Results on validation and test sets.

| Model | Accuracy | PPV | TPR | F1 score |
|---|---|---|---|---|
| RF | 0.845 | 0.848 | 0.827 | 0.837 |
| SVM | 0.887 | 0.88 | 0.887 | 0.883 |
| **Our method** | **0.917** | **0.928** | **0.906** | **0.917** |

## 4. Conclusion

In this work, we propose a deep learning method to encode facial action unit information to recognize hypomimia with PD based on GNN. We demonstrate that using GNN to extract AUs can better represent facial features and their relationships, leading to improve accuracy of hypomimia identification. Through short videos, it can help ordinary users to get more convenient diagnosis. For future work, integrating the characteristics of the disease into the graph construction can increase the medical interpretability of the model, and increase the reliability of Parkinson's disease recognition.

## Acknowledgments

This research was supported in part by the National Key Research and Development Program of China under Grant 2022YFF0902004, and in part by the "Pioneer" and "Leading Goose" R&D Program of Zhejiang under Grant 2023C03101.

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
