# OpenReview forum: "Facial AU-aid hypomimia diagnosis based on GNN"
_MIDL.io/2023/Short_Paper_Track — MIDL 2023 Short paper track Poster_

### Official Review · Reviewer_BRBa · 2023-04-22
**Encouraging results but description and evaluation lacking.**

**Rating:** 5
**Confidence:** 4

**Review:**

This proposes to build a graph on action units extracted from a SwinTR backbone, then use a GNN to predict hypomimia in PD patients.

The idea of building a knn graph from AU features already appears in Chen et al's CVPR 2020 paper 'label distribution learning...', so the novelty here is mainly in 1) using a GNN on top of this graph 2) for this particular diagnosis

Pros:
- Performance looks promising

Cons:
- The selection and representation of AUs is not clear. AUs, which can be computed automatically using tools such as CERT, typically have a clear semantic, e.g. AU45 is eye closure, AU9 is nose wrinkle etc. Is this the case here that the extracted AUs are interpretable? The text says they are non-related, but how is this established? Does it just mean they are extracted, but we don't know the relationships (e.g. Euclidean distances) between them yet?
- Unclear how the AUs serve as node features. This impacts the understanding of the graph construction. What is the dimension of AUs after global average pooling? Is this pooling in time, space, or both? Are AUs vectors?
-  It is unclear if the performance of the proposed architecture derives mainly from using a GNN, that is, essentially exploiting proximity of AUs as a guide to message-passing and aggregation operations, or if it derives from non-linearity. If the 8 AUs are always extracted in the same order somehow, I would suggest just trying an MLP on the AUs, with no graph.
- While the GNN used a validation set, there is no mention of hyperparameter tuning for the SVM or RF. Is the SVM kernel linear? RBF? What is the regularisation (L1? L2?), loss(L1? L2?), and regularisation strength ? This can change performance importantly.

---

### Official Review · Reviewer_W5hj · 2023-04-24
**Pipeline of existing methods, novelty in application and potential of discussions is good.**

**Rating:** 7
**Confidence:** 5

**Review:**

101 Facial AU-aid hypomimia diagnosis based on GNN

This abstract summarizes a new method to extract video features for improving the recognition of hypomimia in parkinsonians. The method consists of a pipeline of existing methods, evaluation indicates improvements over comparative methods. Results could include more detail, bibliography is borderline insufficient. Novelty in application and potential of discussion is good. Recommendation towards Borderline Acceptance.